# Evidence linking atopy and staphylococcal superantigens to the pathogenesis of lymphomatoid papulosis, a recurrent CD30+ cutaneous lymphoproliferative disorder

**Marshall E. Kadin** [1,2,3] *, **Robert G. Hamilton** [4,5,6], **Eric C. Vonderheid** [7,8]

**1** Department of Dermatology, Boston University, Boston, MA, United States of America, **2** Department of Pathology and Laboratory Medicine, Lifespan-Rhode Island Hospital, Providence, RI, United States of America, **3** Department of Dermatology and Skin Surgery, Roger Williams Medical Center, Providence, RI, United States of America, **4** Department of Medicine, Johns Hopkins University School of Medicine, Baltimore, MD, United States of America, **5** Department of Pathology, Johns Hopkins University School of Medicine, Baltimore, MD, United States of America, **6** Johns Hopkins Dermatology, Allergy and Clinical Immunology Reference Laboratory, Baltimore, MD, United States of America, **7** Department of Oncology, Johns Hopkins University School of Medicine, Baltimore, MD, United States of America, **8** Sydney Kimmel Cancer Center, Johns Hopkins University School of Medicine, Baltimore, MD, United States of America

* mkadin@me.com

**Data Availability Statement:** All relevant data are within the manuscript and its Supporting Information files.

## Abstract

### Background

Primary cutaneous CD30+ lymphoproliferative disorders (CD30CLPD) are the second most common type of cutaneous T cell lymphoma (CTCL) and include lymphomatoid papulosis (LyP) and primary cutaneous anaplastic large cell lymphoma (pcALCL). Case reports and small patient series suggest an association of CD30CLPD with atopic disorders. However, the prevalence of atopy in patients with CD30CLPD in retrospective studies depends on patients' recall which is not always reliable. More objective criteria of atopy include evidence of skin reactivity to allergens (positive prick test) and evidence of allergen-specific IgE in serum. This study was undertaken to test the hypothesis that atopy is prevalent in patients with CD30CLPD using serologic criteria of allergen-specific IgE antibodies to aeroallergens and *Staphylococcal aureus* enterotoxin superantigens (SSAgs).

### Methods

We tested serum samples of CD30CLPD for common IgE-specific airborne allergens with the Phadiatop test, which if positive, is regarded as serologic evidence of atopy in adults. Sera were also tested for IgE antibodies reactive to three Staphylococcal enterotoxins with superantigenic properties (SSAg-IgE). Control sera were obtained from adult subjects evaluated for rhino-sinusitis and a negative Phadiatop test. Patients' history of an atopic disorder was obtained by retrospective chart review.

### Findings

Nearly 50% of patients with the most common LyP types (A and C) had a positive Phadiatop test for allergic sensitization to common airborne allergens, and total serum IgE (IgE-t) was

**Funding:** MEK received a grant from the Drs. Martin and Dorothy Spatz Foundation, Sebastopol, CA, to support this work.

**Competing interests:** The authors have declared that no competing interests exist.

increased compared to non-atopic controls. At the IgE antibody concentration generally used to define serologic atopy ($\geq 0.35$ kU$_A$/L), 8/31 (26%) samples of CD30CLPD and 7/28 (25%) samples of LyP were reactive to at least one SSAg-IgE compared to 3/52 (6%) control specimens (P = 0.016 and P = 0.028, respectively). TSST1-IgE was detected in 7 (23%) specimens of CD30CLPD, often together with SEB-IgE; SEA-IgE $\geq 0.35$ kU$_A$/L was not detected. For control specimens, TSST1-IgE exceeded the 0.35 kU$_A$/L threshold in 3 (6%) specimens.

## Conclusions

Patients with LyP types A and C have serologic evidence of atopy against common airborne antigens and SSAgs when compared to control adult subjects who had rhino-sinusitis and a negative Phadiatop test for aero-IgEs. Serologic evidence of atopy exceeded that determined by LyP patients' personal history. The findings support our hypothesis that an atopic diathesis may contribute to the pathogenesis of the most common types of LyP (A and C).

## Introduction

Primary cutaneous CD30+ lymphoproliferative disorder (CD30CLPD) consists of lymphomatoid papulosis (LyP) and primary cutaneous anaplastic large cell lymphoma (pcALCL) at benign and malignant ends of the spectrum, respectively.[1] LyP is characterized clinically by spontaneously regressing papules and nodules (usually less than 2 cm diameter) and is divided into five subtypes A to E based on histo-immunopathologic findings.

The dermal infiltrate of LyP-A, the most common expression of LyP, consists of scattered large CD30+CD4+ cells together with lymphocytes and other inflammatory cells (neutrophils and eosinophils). LyP-B has histopathologic features resembling mycosis fungoides with atypical small-intermediate sized lymphocytes with cerebriform nuclei that usually do not express CD30. LyP-C has a dermal infiltrate with large clusters or sheets of atypical CD30+ cells that might occur in pcALCL. The two other uncommon subtypes of LyP are CD8+ rather than the more typical CD4+ variants. LyP-D is characterized by a dense pagetoid infiltrate of the epidermis by atypical cells that express a CD3+CD4-CD8+ phenotype (rarely CD3+CD4-CD8- phenotype) and CD30 to a variable degree. The dermal infiltrate also contains atypical CD8 + cells. LyP-E is a rare angiocentric variant with CD30+CD8+ cells. pcALCL is defined clinically by large nodules, plaques or tumors that tend to persist although spontaneous regression can occur in up to 40% of lesions. The dermal infiltrate typically contains sheets of large CD30 + cells with or without other inflammatory cells. However, some cases of clinical pcALCL have a dermal infiltrate more typical of LyP-A. Such cases have been designated grade III pcALCL to distinguish them from cases with typical pcALCL (grade IV). Thus, there is an overlap between LyP-A and grade III pcALCL and LyP-C and grade IV pc ALCL.

In a previous study, we reported that IgE-t serum levels of patients diagnosed with CD30CLPD are increased compared to published series of non-atopic patients, and that 30% of patients (30% with LyP) had IgE-t levels greater than 100 kU/L, a threshold commonly used to signify "probable atopy" in adults.[2] However, IgE-t levels did not correlate with the patients' personal or family history of an atopic condition. This observation suggested that factors other than genetic predisposition for atopy (atopic diathesis) of patients mediated the increase in IgE-t, and we hypothesized that atypical CD30+ cells might be directly promoting

IgE production, possibly by secretion of IgE-enhancing cytokines such as IL-13. [3, 4] Although CD30+ cells in LyP often lack expression of the T cell receptor (TCR), [5] CD30 signaling in the absence of the TCR selectively induces IL-13 production. [4] IgE-t levels may be influenced by other factors including age (IgE-t levels in pre-adults > adults), gender (IgE levels in men > women), race (IgE levels in non-White > White), use of tobacco (IgE-t levels in smokers > non-smokers), socioeconomic status, and disease-related factors including bacterial colonization with production of superantigens (SAg).[6]

In this study, we re-evaluate the role of atopy and its association with IgE-t levels by screening available serum samples of CD30CLPD for common IgE-specific airborne allergens (aero-IgE) with the Phadiatop test, which if positive, is regarded as serologic evidence of atopy.[7] We also test sera for IgE reactive against three Staphylococcal enterotoxins with superantigenic properties (SSAg-IgE). Our principal findings are that LyP patients have a significant increase in serologic evidence of atopy with respect to common aeroallergens and at least one SSAg. Serologic evidence of atopy significantly exceeded that of patients' personal history of atopic conditions. The serologic findings suggest that patients' personal history of atopy underestimates the potential contribution of atopy to the pathogenesis of LyP.

## Methods

This study was approved by Lifespan IRB, approval number [955870–7] and Roger Williams Medical Center IRB, approval number [19-520-99], both in Providence RI. The study was conducted in accordance with the Declaration of Helsinki Principles. IgE measurements were performed on de-identified frozen sera. Clinical data were obtained from a cutaneous lymphoma registry approved by the Institutional Review Board at Johns Hopkins University. The ethics committee waived the requirement for informed consent. The CD30CLPD study population consisted of 19 patients with LyP-A (7 men, 12 women; median age, 56 years, range 13 to 77 years), 9 patients with LyP-C (5 men, 4 women; median age, 47 years, range 15 to 80 years), and 3 male patients with pcALCL (median age, 54 years, range 19 to 62 years). All but 2 patients were White. The percentage of CD30+ cells in the dermal infiltrate of the skin specimen obtained at the time of evaluation was visually estimated and categorized into 4 groups: < 5%, 5–19%, 20–49% and $\geq$ 50% CD30+ cells.

The ImmunoCAP250 method (ImmunoCAP, Thermofisher Scientific, Uppsala, Sweden) was used to measure IgE-t and allergen-specific IgEs on de-identified frozen sera collected at the time of clinical evaluation. The Phadiatop multiallergen test was used to screen for IgE reactivity against common airborne allergens in the Northeastern United States. IgEs-specific for Staphylococcal enterotoxin superantigens A,and B (SEA, SEB) and toxic shock syndrome toxin-1 (TSST1) were measured separately. For this analysis, a positive test for aero-IgE or SSAg-IgE was defined as a concentration of $\geq 0.35$ kU$_A$/L, where 1 IU = ~2.4 ng of IgE. The lower limit of detection (LLD) of the assay was 2.0 kU/L for IgE-t and 0.1 kU$_A$/L for allergen-specific IgEs. For values less than the LLD, a fill value of LLD/$\sqrt{2}$ was used for calculation purposes, i.e., 1.41 kU/L for IgE-t and 0.07 kU$_A$/L for allergen-specific IgEs. [6] Sera from adult patients (22 men, 30 women; ages, 24 to 78 years), who were evaluated for rhino-sinusitis and had a negative Phadiatop test for aero-IgEs, were used as a serologic non-atopic control group. Measurement of soluble CD30 (sCD30) in some CD30CLPD patients was performed at ARUP laboratories (Salt Lake City, UT) as part of a previously reported study of cytokine expression.[8]

## Statistics

Results of IgE measurements are reported as median value with a range. Because IgE-t and allergen-specific IgE values are skewed,[6] the distribution of these variables was normalized

by log transformation and the geometric mean (GM), i.e., the anti-log of the mean log10 IgE-t value and its 95% confidence interval (CI), are also provided [6]. The nonparametric Kruskal–Wallis ranks test was used to test differences of median IgE values among independent groups. In addition, log transformed IgE values were used to compare mean values using parametric tests (t-test with equal variances not assumed, one way analysis of variance and Dunnett's t-test for comparison against the control sera). Fisher's and Pearson's chi-square exact tests were used to test categorical data in 2 by 2 and R by C tables, respectively. Spearman's rank correlation coefficient was used to determine the strength of association between IgE-t and sCD30 values. The statistical software used in the study were SYSTAT10 and SPSS 13.0 for Windows, SPSS, Inc. (Chicago, IL) and StatXact-3, Cytel, Inc. (Cambridge, MA).

# Results

## IgE-t is increased in patients with LyP but not pcALCL

Table 1 summarizes the results of IgE-t according to diagnostic categories in this study. In agreement with our previous study, [2] IgE-t values for LyP-A and LyP-C subgroups were significantly higher than non-atopic controls. However, IgE-t was not increased for pcALCL, but this was based on only 3 patients. To clarify this issue, IgE-t values from reference labs were analyzed on 16 additional patients with pcALCL not included in this study. The IgE-t GM was 32.0 kU/L (CI, 13.9–73.7 kU/L); this was still lower than the GM of 77.5 kU/l (CI, 45.8–131) for patients with LyP-A+C (P = 0.251).

## Clinical atopy and aero-IgE screen (Phadiatop test)

Fifteen of 31 (48%) patients with CD30CLPD had a positive ($\geq$ 0.35 kU/L) Phadiatop multi-allergen test (S1 Table). Only 2 of these patients (13%) had a personal history of an atopic disorder (both with allergic rhinitis). Conversely, 5 of 16 (31%) with a negative Phadiatop test had a history of atopy (4 patients with allergic rhinitis, one with childhood atopic dermatitis). These differences were not statistically significant (P = 0.394) Similar results were found for patients with LyP-A and LyP-C. These findings suggest that the prevalence of serologic atopy is higher than the patients' personal recall of atopic conditions for this cohort of patients.

The lack of a correlation between patients' history and serologic evidence of atopy for this subset of patients prompted an investigation of an additional 105 patients with CD30CLPD who were not studied for specific aero-IgEs (S2 Table). A personal history of an atopic condition was documented in the medical records of 41 (39.0%) of patients overall. For 82 patients diagnosed with LyP-A and LyP-C, an atopic condition was recorded for 35 (43%) patients

**Table 1. Total serum IgE according to diagnostic categories.**

| Diagnosis | No. | Median (range) | GM (95% CI) | KW† | t† |
|---|---|---|---|---|---|
| All CD30CLPD | 31 | 53.7 (4.3–927) | 66.8 (39.5–113) | < 0.001 | < 0.001 |
| All LyP | 28 | 58.2 (6.6–927) | 77.5 (45.8–131) | < 0.001 | < 0.001 |
| LyP-A | 19 | 62.6 (6.6–927) | 79.2 (39.5–159) | < 0.001 | < 0.001 |
| LyP-C | 9 | 53.7 (11.7–912) | 74.1 (28.8–190) | 0.001 | 0.003 |
| pcALCL | 3 | 10.4 (4.3–104) | 16.7 (0.3–993) | 0.956 | 0.878 |
| Controls | 52 | 14.8 (2–187) | 14.1 (10.1–19.9) | ---- | ---- |

Abbreviations: CD30CLPD, primary cutaneous CD30+ lymphoproliferative disease; LyP, lymphomatoid papulosis. GM, geometric mean and 95% confidence interval
† P-values for disease group versus control: KW, Kruskal-Wallis and t tests of 2 independent samples.
The difference in IgE-t levels between LyP-A and LyP-C was not significant (KW, P = 0.863; t test, 0.900).

which was higher than the 21% prevalence for the current cohort of LyP patients (P = 0.069). The difference was apparent for patients with LyP-A (40% vs. 11%, P = 0.024), but not LyP-C (44% vs. 50%, P = 1.0). Furthermore, the supplemental patients' personal history of an atopic condition was associated with increased levels of IgE-t for patients with LyP-A, but not LyP-C (S3 Table). Of interest, IgE-t was also increased for LyP-A patients with a history of penicillin allergy compared to patients without penicillin allergy (not shown). These observations suggest that a link may exist between patient's atopic diathesis and the development of LyP-A.

## Serologic evidence of atopy and IgE-t levels

Sixteen of 28 (57%) patients with LyP had no serologic evidence of atopy (aero-IgE < 0.35 $kU_A/L$). The IgE-t value for these patients (median, 53.5 kU/L, range, 4.3–927 kU/L; GM, 55.4 kU/L (CI, 22.4–137 kU/L) was significantly higher than the control group (KW, P = 0.001; t-test, P = 0.002; Table 2). This finding suggests that some factor other than the patients' atopic diathesis might be influencing IgE production.

## Relationship between CD30+ cells in skin lesions and IgE-t

In our previous study, [2] IgE-t GM values progressively increased for cases classified from LyP-B to pcALCL; however, the differences were not statistically significant. Given that the number of atypical CD30+ cells in the dermis generally increases from LyP-B to LyP-A to LyP-C to pcALCL, this observation suggested that CD30+ cells in skin lesions might be mediating IgE-t production. However, for the current series, IgE-t levels were about the same for LyP-A (GM, 79.2 kU/L) and LyP-C (GM, 74.1 kU/L) and lower for 3 cases of pcALCL (GM, 16.7 kU/L; Table 1).

We therefore re-evaluated the relationship between diagnosis and IgE-t in 105 additional patients with CD30CLPD and, for comparison, 16 patients with pityriasis lichenoides, a closely related cutaneous lymphoproliferative disorder (S1 Fig) and (S4 Table). The GMs for previously measured IgE-t (restudied cases excluded) again increased from LyP-B to LyP-C, but then decreased for patients with pcALCL. However, these differences again were not statistically significant (KW, P = 0.570; ANOVA, P = 0.474).

Of interest, IgE-t values for patients with LyP-D, which expresses a CD8+ phenotype, were quite low (GM, 17.3 kU/L) and IgE-t values for pityriasis lichenoides (GM, 43.6 kU/L) were intermediate between LyP-B and LyP-A.

In addition, the estimated number of atypical CD30+ cells in the dermal infiltrate of skin specimens taken at the time of blood sampling was correlated to IgE-t levels. No correlation was found among categorized groups and IgE-t values S5 Table). Moreover, no correlation was found between the level of soluble CD30 (sCD30) in the blood and IgE-t for all previously studied CD30CLPD patients (n = 76, rho, 0.149, P = 0.198; (S2 Fig) and LyP-A only (n = 44, rho = 0.106, P = 0.492).

## Prevalence of Staphylococcal superantigen-specific IgE

Measurable IgE antibody reactivity ($\geq$ 0.1 $kU_A/L$) against at least one SSAg was present in 12/31 (39%) specimens of CD30CLPD compared to 6/52 (12%) non-atopic control specimens (P = 0.006; Table 3). At the concentration used to define serologic atopy with aero-IgEs ($\geq$ 0.35 $kU_A/L$), 8/31 (26%), samples of CD30CLPD and 7/28 (25%) samples of LyP remained positive compared to 3/52 (6%) control specimens (P = 0.016 and P = 0.028, respectively). At this threshold, TSST1-IgE was detected in 7 (23%) specimens of CD30CLPD, often together with SEB-IgE (6 specimens). SEA-IgE $\geq$ 0.35 $kU_A/L$ was not detected. For control specimens, TSST1-IgE exceeded the 0.35 $kU_A/L$ threshold in 3 specimens (0.43, 0.47 and 0.93 $kU_A/L$).

**Table 2. Relationship between presence or absence of airborne or staphylococcal superantigen specific IgE and serum total IgE for categories of primary cutaneous CD30+ lymphoproliferative disease.**

| Disease Category | Aero-IgE kUa/lL | SSAg-IgE kUa/L | No. In Cohort | Median IgE-t (range) kU/L | GM (95% CI) kU/L |
|---|---|---|---|---|---|
| CD30CLPD | < 0.35 | < 0.35 | 12 | 28.8 (4.3–420) | 29.9 (12.5–71.3) |
| | < 0.35 | ≥ 0.35 | 4 | 587 (70.8–927) | 354 (50.3–2485) |
| | ≥ 0.35 | < 0.35 | 11 | 47.5 (20.7–598) | 66.3 (32.8–134) |
| | ≥ 0.35 | ≥ 0.35 | 4 | 180 (33.4–491) | 145 (22.6–924) |
| | < 0.35 | Any | 16 | 53.5 (4.3–927) | 55.4 (22.4–137) |
| | Any | < 0.35 | 23 | 44.1 (4.3–598) | 43.7 (24.3–75.6) |
| LyP-A+C | < 0.35 | < 0.35 | 10 | 39.7 (6.6–420) | 40.3 (16.1–101) |
| | < 0.35 | ≥ 0.35 | 4 | 587 (70.8–927) | 354 (50.3–2485) |
| | ≥ 0.35 | < 0.35 | 11 | 47.5 (20.7–598) | 66.3 (32.8–134) |
| | ≥ 0.35 | ≥ 0.35 | 3 | 256 (33.4–491) | 161.3 (5.0–5253) |
| | < 0.35 | Any | 14 | 66.8 (6.6–927) | 75.0 (29.9–188) |
| | Any | < 0.35 | 21 | 46.7 (6.6–598) | 52.3 (30.8–88.8) |
| LyP-A | < 0.35 | < 0.35 | 6 | 28.8 (6.6–420) | 40.9 (7.7–219) |
| | < 0.35 | ≥ 0.35 | 3 | 261 (70.8–927) | 258 (10.6–6291) |
| | ≥ 0.35 | < 0.35 | 8 | 50.5 (20.7–598) | 74.1 (26.5–207) |
| | ≥ 0.35 | ≥ 0.35 | 2 | 262 (33.4–491) | 128 |
| | < 0.35 | Any | 9 | 70.8 (6.6–927) | 75.6 (20.6–277) |
| | Any | < 0.35 | 14 | 36.8 (6.6–598) | |
| LyP-C | < 0.35 | < 0.35 | 4 | 53.5 (11.7–74) | 39.4 (10.4–150) |
| | < 0.35 | ≥ 0.35 | 1 | 912 | ---- |
| | ≥ 0.35 | < 0.35 | 3 | 47.5 (46.7–53.7) | 49.2 (40.7–59.5) |
| | ≥ 0.35 | ≥ 0.35 | 1 | 128 | ---- |
| | < 0.35 | Any | 5 | 62.8 (11.7–912) | 73.9 (20.6–277) |
| | Any | < 0.35 | 7 | 47.5 (11.7–74.4) | 1.3 (24.7–75.9) |
| Control | < 0.35 | < 0.35 | 49 | 13.5 (2.0–110) | 12.5 (9.0–17.3) |
| | < 0.35 | ≥ 0.35 | 3 | 132 (56.0–187) | 111 (23.9–520) |
| | < 0.35 | Any | 52 | 14.8 (2.0–187) | 14.1 (10.1–19.9) |

Abbreviations: CD30CLPD, primary cutaneous CD30+ lymphoproliferative disease; LyP, lymphomatoid papulosis. GM, geometric mean and 95% confidence interval.

Comparison IgE-t values for subgroups with < 0.35 kUa/L aero-IgE and < 0.35 kUa/L SSAg-IgE:

IgE-t for the 3 CD30CLPD categories (KW, P = 149 and ANOVA, P = 0.257)

IgE-t for LyP-A versus LyP-C subgroups (KW, P = 0.831 and t-test, P = 0.963)

Comparison IgE-t values for diagnostic categories with < 0.35 kUa/L aero-IgE and < 0.35 kUa/L SSAg-IgE versus controls: KW, P = 0.065; ANOVA, P = 0.033

All CD30CLPD versus control: KW, P = 0.065; t-test, P = 0.059

All LyP versus control: KW, P = 0.017; t-test, P = 0.020

LyP-A only versus control: KW, P = 0.108; t-test, P = 0.131

LyP-C only versus control: KW, P = 0.047; t-test, P = 0.063

pcALCL only versus control: KW, P = 0.369; t-test, P = 0.373

The difference in frequency for TSST1-IgE alone remained significantly higher for all CD30CLPD samples (P = 0.002) and marginal for the LyP subset (P = 0.059).

## Relationship between SSAg-IgE and CD30 expression in skin specimens

The possibility that SSAgs might play a role in the proliferation of CD30+ cells of LyP was investigated by correlating the presence of SSAg-IgE with type of CD30CLPD and level of

**Table 3. Distribution of positive Staphlococcal superantigen-specific IgE at two detection thresholds ($\geq$ 0.1 k$_A$U/L and $\geq$ 0.35 kU$_A$/L) according to disease categories.**

| SSAg-IgE ($\geq$ 0.1 kUa/L) | CD30CLPD N = 31 | LyP-A+C N = 28 | Control N = 52 |
|---|---|---|---|
| TSST1 alone | 5 | 5 | 3 |
| TSST1+SEA | 0 | 0 | 2 |
| TSST1+SEB | 4 | 3 | 0 |
| TSST1+SEA+SEB | 2 | 2 | 0 |
| SEA alone | 0 | 0 | 0 |
| SEA+SEB | 1 | 1 | 0 |
| SEB alone | 0 | 0 | 1 |
| Any SSAg | 12 (39%) | 11 (39%) | 6 (12%) |
| P-value† | 0.006 | 0.008 | - - - - |
| SSAg-IGE ($\geq$ 0.35 kUa/L) | CD30CLPD N = 31 | LyP-A+C N = 28 | Control N = 52 |
| TSST1 lone | 1 | 1 | 2 |
| TSST1+SEA | 0 | 0 | 1 |
| TSST1+SEB | 4 | 3 | 0 |
| TSST1+SEA+SEB | 2 | 2 | 0 |
| SEA alone | 0 | 0 | 0 |
| SEA+SEB | 1 | 1 | 0 |
| SEB alone | 0 | 0 | 0 |
| Any SSAg | 8 (26%) | 7 (25%) | 3 (6%) |
| P-value† | 0.016 | 0.028 | - - - - |

† Frequency of positive test, disease versus control groups, Fisher's exact test

CD30 expression in the skin specimen obtained at the time of blood sampling. Samples with SSAg-IgE values $\geq$ 0.1 kU$_A$/L or $\geq$ 0.35 kU$_A$/L did not occur more often for patients with higher numbers of CD30+ cells in the skin versus lower numbers, i.e., LyP-C/pcALCL vs. LyP-A or CD30+ dermal cells $\geq$ 20% vs. < 20%; S6 Table).

## Relationship between SSAg-IgE and serologic atopy

The relationship between serologic atopy, SSAg-IgE and IgE-t for each disease category is shown in Table 2. To minimize the influence of atopic diathesis and SSAgs on IgE-t levels, IgE-t levels of patients without a positive test for either aero-IgE and SSAg-IgE (< 0.35 kU$_A$/L) were compared against non-atopic controls (Table 2). Although the numbers were small, IgE-t levels for patients with LyP were significantly higher than controls. No difference in IgE-t levels was apparent among the diagnostic categories. These observations suggest that additional factor or factors apart from the patients' atopic diathesis and perhaps presence of CD30+ cells are influencing IgE-t levels.

Table 2 also indicates that patients with CD30CLPD who are sero-negative for aero-IgEs but positive for SSAg-IgE have higher IgE-t levels compared to patients who are sero-negative to both aero-IgE and SSAg-IgE (P = 0.020). The difference was statistically significant for all LyP patients (P = 0.032), but not LyP-A alone (P = 0.122), perhaps because of the small number of patients.

Likewise, IgE-t levels were higher in patients with CD30CLPD or LyP who are sero-positive for aero-IgEs, but negative for SSAg-IgE. However, the differences were not statistically significant (P = 0.131 and P = 0.348, respectively). Consequently, IgE-t values were also high for the small number of patients who were sero-positive for both aero-IgE and SSAg-IgE.

## Systemic corticosteroid usage and IgE-t levels

Systemic corticosteroids might suppress production of IgE. However, IgE-t levels were significantly higher for 5 LyP-A patients who were taking systemic corticosteroids (10 to 60 mg prednisone per day) at the time of study (S7 Table). One patient had corticosteroid responsive eczematous dermatitis co-existing with LyP-A. We attribute the positive correlation between systemic corticosteroid usage and IgE-t to disease severity rather than an IgE-enhancing effect by corticosteroids.

## Smoking and IgE-t

Some studies have shown increased IgE-t in people who smoke. [6] However, in accordance with our previous study, IgE-t was not increased in patients with CD30CLPD [2] (S8 Table).

## Discussion

In this study, 50% of patients with LyP types A and C had serologic evidence of atopy (positive Phadiatop multiallergen screen), and IgE-t was increased compared to non-atopic controls. As in our previous study, [2] no correlation was apparent between serologic atopy and the patients' history of an atopic condition for this small cohort of patients. Nevertheless, a review of medical records of an additional 105 patients with CD30CLPD showed that IgE-t levels were increased in patients with a personal atopic history compared to patients without clinical atopy. This association was statistically significant only for LyP-A, and not for LyP-C, pcALCL or pityriasis lichenoides.

These results suggest that a pathogenic link may exist between an atopic diathesis and development of LyP-A and perhaps LyP-C. Accordingly, Nijsten noted an atopic condition in 18 of 35 (51%) cases of childhood LyP (atopic dermatitis in 2 patients, allergic rhinitis in 12 patients, allergic asthma in 3 patients, and allergic rhinitis plus asthma in 1 patient), and Miquel reported 7 of 25 (28%) children with LyP had atopic dermatitis. [9, 10] In addition, Fletcher and Laube each reported an adult patient with LyP-A and active atopic dermatitis, [11, 12] and in our series, we encountered 2 adults with long-standing atopic dermatitis and LyP (one LyP-A, one LyP-C) and several other patients with non-specific eczema. Although not observed by us, several cases of pcALCL occurring in patients with atopic dermatitis have been reported, often in the context of cyclosporine therapy given for the atopic dermatitis. [13, 14]

In agreement with our previous study, [2] IgE-t GM values of supplemental patients progressively increased from LyP-B to LyP-A to LyP-C, and with increasing categories of CD30 + dermal cells below 50% in the skin specimen obtained at the time of blood sampling. This observation suggests that the atypical CD30+ cells might be influencing IgE production. Indeed, studies in the Kadin laboratory have shown that CD30+ large atypical cells of LyP produce IL-13 which is an IgE-enhancing cytokine (Fig 1). However, if CD30+ cells directly influence IgE-t production, then the decrease in IgE-t for patients diagnosed to have pcALCL compared to LyP-C requires explanation. One possibility is that the total biomass of CD30 + cells in pcALCL may be less than LyP-C. This might occur because pcALCL is characterized by fewer skin lesions than those of patients with LyP-C, and the infiltrate of some pcALCL lesions may have fewer than 50% CD30+ cells, thereby resembling LyP-A.[15] Nevertheless, the lack of a correlation between IgE-t and sCD30 blood levels suggests that CD30+ cells per se are not the major factor mediating IgE-t levels.

What factors other than CD30+ cells might account for the increased IgE in LyP? In addition to atopic diathesis, SSAgs may be directly involved in stimulating IgE production given that IgE-t levels were higher for non-atopic patients with SSAg-IgE levels $\geq 0.35$ kU$_A$/L than

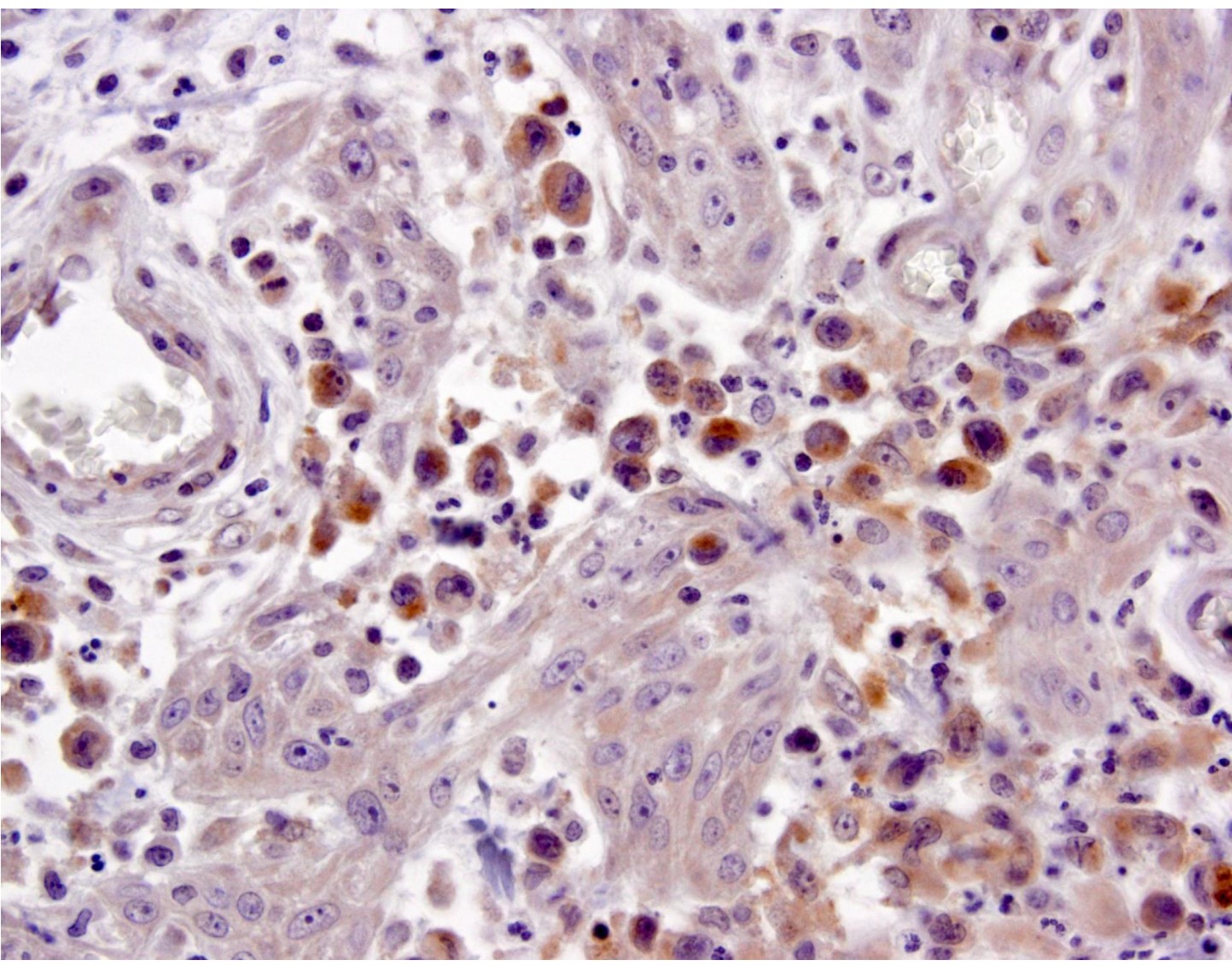

**Fig 1. Large atypical cells of LyP-A express IL-13.** Keratinocytes and granulocytes are unstained. Immunohistochemistry was done on formalin-fixed paraffin-embedded tissue with a mouse anti-human IL-13 monoclonal antibody (1microgram/ml) from Abcam (Cambridge, UK).

patients with $< 0.35$ kU$_A$/L (Table 2). Patients' age and gender, smoking history and use of systemic corticosteroids are seemingly not important.

The detection of SSAg-IgEs in nearly 50% of patients with LyP indicates that exposure to bacterial superantigens, particularly TSST1, has taken place. One possibility is that SSAg-producing S. aureus gain entry into ulcerated LyP lesions and that the resultant production of SSAg-IgE was exaggerated in patients with an atopic diathesis. For example, it seems possible that S. aureus might activate skin resident type 2 innate lymphoid cells (ILC2) which then secrete IL-5 and IL-13, express CD40L (CD154) and interact with CD40+ B-cells to increase IgE including SSAg-IgE. To account for the high IgE-t levels in LyP types A and perhaps C, LyP patients with an atopic diathesis have hyper-responsive and/or increased numbers of ILC2 cells in the skin compared to non-atopic individuals as has been reported for atopic dermatitis. [16] [17] [18] The relationship between atopic diathesis and ILC2 would also explain

why IgE-t is often increased in a variety of other inflammatory skin diseases.[2] Further studies are required to test this hypothesis.

An alternative possibility is that SSAgs have a direct pathogenic role similar to what has been proposed for eruptive guttate psoriasis.[19, 20] This would explain the eruptive nature of LyP. According to this hypothesis, SSAgs from S. aureus (or other microorganism) activate precursor neoplastic CD4+ and occasionally CD8+ T cells that then express CLA and other skin-homing molecules. After entry into the dermis, these cells proliferate with expression of CD30. Spontaneous regression may be mediated by TGF-beta that is secreted by these cells. [21] A similar mechanism might be proposed for the closely related pityriasis lichenoides.[22]

In patients with eruptive guttate psoriasis, tonsils are the usual source of Streptococcal bacteria that elaborate TSST1 superantigen.[23] In patients with LyP, the bacterial source could also be an extracutaneous location such as the nasal mucosa which is frequently colonized by S. aureus, particularly patients who are prone to atopy. [24] If SSAgs mediate the initial proliferation of the atypical cells of LyP, then CD30+ cells might have a skewed TCR-Vbeta profile, e.g., increased TCR-Vbeta 2 expression which corresponds to stimulation by TSST1 superantigen. Thus far, data to support or refute this hypothesis are limited.[25]

Current studies indicate that colonized SSAg-producing S. aureus intensify cutaneous inflammation of atopic dermatitis and Sézary syndrome and promote tumor progression in Sézary syndrome. [26, 27] Antibiotics have an ameliorating effect on inflammation in these conditions, [28] [29]

Oral antibiotics are also administered as a treatment for LyP. However, because LyP is a chronic condition characterized by intermittent flare ups of lesions, it is difficult to determine the long term effectiveness of anti-Staphylococcal antibiotics for LyP. To our knowledge, formal studies have not been performed. In Dr. Vonderheid's experience, long term administration of tetracycline (no longer available) or minocycline in doses used to treat acne vulgaris can often ameliorate LyP by suppressing new lesion formation. The underlying mechanism for this may be anti-inflammatory as well as anti-bacterial. In this regard, the observation by Alexander-Savino that doxycycline inhibits neoplastic T cell growth by acting as an NF-kappaB inhibitor provides another potential mechanism for the beneficial effect of tetracycline-type drugs for LyP. [30] A clinical trial to determine if long term administration of doxycycline is useful as a suppressive treatment for LyP might be worth considering.

A limitation of this study is the relatively small number of patients for whom sera and corresponding clinical histories were available, largely due to the rarity of CD30CLPD. A second limitation is analysis of only 3 SSAgs as a larger number SSAgs might disclose further evidence of IgE specific antibodies.

In summary, patients with the most common LyP types A and C have serologic evidence of atopy against common airborne antigens and SSAgs when compared to control adult subjects who had rhino-sinusitis and a negative Phadiatop test for aero-IgEs. Serologic evidence of atopy exceeded that determined by LyP patients' personal history. The results suggest that an atopic diathesis and perhaps staphylococcal superantigens contribute to the pathogenesis of LyP.

## Supporting information

**S1 Fig. Serum total IgE values for supplemental patients with primary cutaneous CD30+ lymphoproliferative disorder or pityriasis lichenoides (PL). The GM and 95% confidence interval of IgE-t are shown for each diagnostic category.** Number patients in parentheses. (DOCX)

**S2 Fig. Correlation between IgE-t level and soluble CD30 level in the blood of patients with primary cutaneous CD30+ lymphoproliferative disorder.**
(DOCX)

**S1 Table. Lack of correlation between patients' history (PHx) of an atopic condition and positive Phadiatop test for atopy (aero-IgE $\geq$ 0.35 kUa/L).**
(DOCX)

**S2 Table. Correlation between supplemental patients' personal history (PHx) of an atopic condition according to disease category and total serum IgE values.**
(DOCX)

**S3 Table. Relationship among disease categories, Phadiatop test results for aero-IgE and total serum IgE.**
(DOCX)

**S4 Table. Relationship between diagnosis and previously measured total serum IgE for 105 supplemental patients with primary cutaneous CD30+ lymphoproliferative disorder and patients with pityriasis lichenoides.**
(DOCX)

**S5 Table. Relationship between number of CD30+ dermal cells and previously measured total serum IgE for supplemental patients with primary cutaneous CD30+ lymphoproliferative disorder.**
(DOCX)

**S6 Table. Correlation SSAg-IgE levels (kUa/L) with subtype of primary cutaneous CD30 + lymphoproliferative disorder and number of CD30+ dermal cells in skin specimens.**
(DOCX)

**S7 Table. Serum total IgE levels according to use of systemic corticosteroids (SCS) for patients with primary cutaneous CD30+ lymphoproliferative disorder.**
(DOCX)

**S8 Table. Total serum IgE levels according to smoking history for patients with primary cutaneous CD30+ lymphoproliferative disorder.**
(DOCX)

## Author Contributions

**Conceptualization:** Marshall E. Kadin, Eric C. Vonderheid.

**Data curation:** Eric C. Vonderheid.

**Funding acquisition:** Marshall E. Kadin.

**Investigation:** Marshall E. Kadin, Robert G. Hamilton, Eric C. Vonderheid.

**Methodology:** Robert G. Hamilton, Eric C. Vonderheid.

**Resources:** Eric C. Vonderheid.

**Validation:** Robert G. Hamilton, Eric C. Vonderheid.

**Writing – original draft:** Marshall E. Kadin.

**Writing – review & editing:** Robert G. Hamilton, Eric C. Vonderheid.

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
