## [Decision Letter · Decision Letter 0]

11 Dec 2019

PONE-D-19-22149

Evidence linking atopy and staphylococcal superantigens to the pathogenesis of lymphomatoid papulosis, a recurrent CD30+ cutaneous lymphoproliferative disorder

PLOS ONE

Dear Professor Kadin,

First of all we would like to apologize for our late response due to difficulties in securing the adequate number of reviewers.

We thank you for submitting your manuscript to PLOS ONE. After careful consideration, the content was considered very interesting and sound. Therefore, we decided yo accept the manuscript for publication provided you agree to address the minor comments raised by one of the reviewers.

We would appreciate receiving your revised manuscript by Jan 25 2020 11:59PM. To enhance the reproducibility of your results, we recommend that if applicable you deposit your laboratory protocols in protocols.io, where a protocol can be assigned its own identifier (DOI) such that it can be cited independently in the future. For instructions see: http://journals.plos.org/plosone/s/submission-guidelines#loc-laboratory-protocols

We look forward to receiving your revised manuscript.

Yours Sincerely,

Lucienne Chatenoud

Academic Editor

PLOS ONE

Journal Requirements:

2. In ethics statement in the manuscript and in the online submission form, please provide additional information about the patient records used in your retrospective study. Specifically, please ensure that you have discussed whether all data were fully anonymized before you accessed them and/or whether the IRB or ethics committee waived the requirement for informed consent. If patients provided informed written consent to have data from their medical records used in research, please include this information.

Additional Editor Comments (if provided):

Reviewers' comments:

Reviewer's Responses to Questions

**Comments to the Author**

1. Is the manuscript technically sound, and do the data support the conclusions?

Reviewer #1: Yes

Reviewer #2: Yes

2. Has the statistical analysis been performed appropriately and rigorously? 

Reviewer #1: I Don't Know

Reviewer #2: Yes

3. Have the authors made all data underlying the findings in their manuscript fully available?

Reviewer #1: Yes

Reviewer #2: Yes

4. Is the manuscript presented in an intelligible fashion and written in standard English?

Reviewer #1: Yes

Reviewer #2: Yes

5. Review Comments to the Author

Reviewer #1: This is an interesting manuscript suggesting that an atopic diathesis may contribute to the pathogenesis of LyP.

If indeed atypical CD30+ cells directly promoting IgE production one should expect that patients with high burden of disease will have higher IgE levels - have you correlated number of LyP lesions with IgE levels?

Reviewer #2: The study is well performed and technically sound with high quality figures and tables. The manuscript is well-read and the conclusions justified. The study builds on previous on previous findings on a putative link between atopy and LyP and provides new interesting data strengthening such an association and a possible role of staphylococcal aureus superantigens in the pathogenesis.

I have no major suggestions for changes of this excellent and interesting study.

A minor point which might be of interest to touch in the Discussion is concerning the role of Staphylococcus aureus superantigens in the pathogenesis. The authors propose a mechanism that SSAg may induce apoptosis in malignant CD30+ T cells in LyP patient and thus hypothetically contribute to the hig.h apoptotic index and benign course of the disease. This is naturally one interesting hypothesis. However, at the same time the authors refer to Dr. Vonderheids observations that antibiotics have a beneficial effect in some patients, suggesting the possibility, that bacteria may a negative clinical effect. Therefore, it could be of interest to know whether or not presence of anti-SSAg of IgE type is associated with regression, progression, or neither (if the authors might have followed these patients for a sufficiently long observation time). Of notice, Lindahl et al (Blood 2019) recently reported that antibiotics inhibited both clinical disease activity and the fraction of malignant T cells in situ in a smal cohort of patients of mycosis fungoides/Sezary syndrome colonised by SSAg producing Staph aureus which is in line with Dr Vonderheids observations and may be worth discussing in relation to LyP.

6. PLOS authors have the option to publish the peer review history of their article (what does this mean?). If published, this will include your full peer review and any attached files.

Reviewer #1: No

Reviewer #2: No

---

## [Author Response · Author response to Decision Letter 0]

21 Jan 2020

Editor’s comment: In ethics statement in the manuscript and in the online submission form, please provide additional information about the patient records used in your retrospective study. Specifically, please ensure that you have discussed whether all data were fully anonymized before you accessed them and/or whether the IRB or ethics committee waived the requirement for informed consent. If patients provided informed written consent to have data from their medical records used in research, please include this information.

Response: This study was conducted in accordance with the Declaration of Helsinki Principles. Clinical data were obtained from a cutaneous lymphoma registry approved by the Institutional Review Board at Johns Hopkins University. IgE measurements were performed on de-identiﬁed frozen sera. The ethics committee waived the requirement for informed consent.

Editor’s comment: We note that you have included the phrase “data not shown” in your manuscript. Unfortunately, this does not meet our data sharing requirements. PLOS does not permit references to inaccessible data. We require that authors provide all relevant data within the paper, Supporting Information files, or in an acceptable, public repository. Please add a citation to support this phrase or upload the data that corresponds with these findings to a stable repository (such as Figshare or Dryad) and provide and URLs, DOIs, or accession numbers that may be used to access these data. Or, if the data are not a core part of the research being presented in your study, we ask that you remove the phrase that refers to these data.

Response: We deleted the sentence ending “data not shown” because it is not essential to our results or conclusion. 

Response to Reviewers

Reviewer #1: This is an interesting manuscript suggesting that an atopic diathesis may contribute to the pathogenesis of LyP.

If indeed atypical CD30+ cells directly promoting IgE production one should expect that patients with high burden of disease will have higher IgE levels - have you correlated number of LyP lesions with IgE levels?

Reply to reviewer 1: Unfortunately, because this was a retrospective study, we cannot provide specific information about the number of LyP lesions or extent of body surface involvement. For this reason, we used soluble CD30 levels from a previous study as measure of the total CD30+ cell mass. As stated in the paper, no correlation was found between the level of sCD30 and IgE-t.

Reviewer #2: The study is well performed and technically sound with high quality figures and tables. The manuscript is well-read and the conclusions justified. The study builds on previous on previous findings on a putative link between atopy and LyP and provides new interesting data strengthening such an association and a possible role of staphylococcal aureus superantigens in the pathogenesis.

I have no major suggestions for changes of this excellent and interesting study.

A minor point which might be of interest to touch in the Discussion is concerning the role of Staphylococcus aureus superantigens in the pathogenesis. The authors propose a mechanism that SSAg may induce apoptosis in malignant CD30+ T cells in LyP patient and thus hypothetically contribute to the hig.h apoptotic index and benign course of the disease. This is naturally one interesting hypothesis. However, at the same time the authors refer to Dr. Vonderheids observations that antibiotics have a beneficial effect in some patients, suggesting the possibility, that bacteria may a negative clinical effect. Therefore, it could be of interest to know whether or not presence of anti-SSAg of IgE type is associated with regression, progression, or neither (if the authors might have followed these patients for a sufficiently long observation time). Of notice, Lindahl et al (Blood 2019) recently reported that antibiotics inhibited both clinical disease activity and the fraction of malignant T cells in situ in a smal cohort of patients of mycosis fungoides/Sezary syndrome colonised by SSAg producing Staph aureus which is in line with Dr Vonderheids observations and may be worth discussing in relation to LyP.

Reply to reviewer 2: We are aware of Lindahl’s work that SSAgs from S aureus intensify skin inflammation and have a tumor promoting effect in patients with mycosis fungoides/Sezary syndrome. However, these patients (as well as atopic dermatitis) have chronic skin lesions that are prone to bacterial colonization. This differs from patients with LyP where lesions come and go, often with intervals where no lesions are present. For this reason, the pathogenic role of SSAg, if any, in LyP may be different. In other words, the eruptive nature of LyP (and the closely related pityriasis lichenoides) more closely resembles eruptive guttate psoriasis than Sezary syndrome. Accordingly we have modified the discussion to clarify this point.

Reviewer 2 ’s comment also refers to this sentence in the discussion: “Although overt clinical infection by S. aureus is not typical of LyP lesions, it is intriguing that LyP sometimes improves on oral antibiotics (particularly tetracycline/minocycline in Dr. Vonderheid’s experience).” If S. aureus is producing SSAgs that stimulate initial expansion if atypical CD30+ cells, then anti-Staphylococcal antibiotics might in principle inhibit the development of new lesions and eventually promote complete clearing. The rationale to administer antibiotics for LyP and Dr. Vonderheid’s experience with long-term tetracycline-type antibiotics as a suppressive therapy has been clarified in the discussion.

---

## [Editor Report · Decision Letter 1]

23 Jan 2020

Evidence linking atopy and staphylococcal superantigens to the pathogenesis of lymphomatoid papulosis, a recurrent CD30+ cutaneous lymphoproliferative disorder

PONE-D-19-22149R1

Dear Dr. Kadin,

We are pleased to inform you that the revised version of your manuscript has been judged scientifically suitable for publication and will be formally accepted for publication once it complies with all outstanding technical requirements.

With best regards,

Lucienne Chatenoud

Academic Editor

PLOS ONE
---

## [Editor Report · Acceptance letter]

3 Feb 2020

PONE-D-19-22149R1 

Evidence linking atopy and staphylococcal superantigens to the pathogenesis of lymphomatoid papulosis, a recurrent CD30+ cutaneous lymphoproliferative disorder 

Dear Dr. Kadin:

I am pleased to inform you that your manuscript has been deemed suitable for publication in PLOS ONE. Congratulations! Your manuscript is now with our production department. 

With kind regards,

on behalf of

Prof. Lucienne Chatenoud 

Academic Editor

PLOS ONE